# Global Profiling of Protein β-hydroxybutyrylome in Porcine Liver

**DOI:** 10.3390/biology14091183

**Published:** 2025-09-02

**Authors:** Shuhao Fan, Jinyu Guan, Fang Tian, Haibo Ye, Qianqian Wang, Lei Lv, Yuanyuan Liu, Xianrui Zheng, Zongjun Yin, Xiaodong Zhang

**Affiliations:** 1College of Animal Science and Technology, Anhui Agricultural University, Hefei 230036, China; fanshuhao@stu.ahau.edu.cn (S.F.); guanjy324@stu.ahau.edu.cn (J.G.); tianfang@stu.ahau.edu.cn (F.T.); yhb010823@stu.ahau.edu.cn (H.Y.); wangqianqian@stu.ahau.edu.cn (Q.W.); lvlei187@stu.ahau.edu.cn (L.L.); zxr07sk1@163.com (X.Z.); yinzongjun@ahau.edu.cn (Z.Y.); 2The First Affiliated Hospital of Anhui University of Chinese Medicine, Hefei 230036, China; zxd1983@163.com

**Keywords:** β-hydroxybutyrylation, metabolic regulation, TCA cycle, ketone bodies, porcine liver

## Abstract

β-hydroxybutyrate (Kbhb), the primary ketone body metabolite, has been identified as an epigenetic regulator capable of modifying proteins. By mapping the Kbhb profile in pig liver, we revealed abundant Kbhb-modified proteins enriched in metabolism-related pathways. Metabolomic analysis further suggested that Kbhb may drive metabolic reprogramming. These findings provide novel insights into the biological functions of Kbhb.

## 1. Introduction

Cells within the same living organism generally share an identical genome. However, they develop in different directions depending on the tissue they are in and the specific stage, and this post-genomic genetic regulatory mechanism is known as epigenetics. Proteins are the executors of life activities. Protein post-translational modifications (PTMs), as a regulatory mechanism at the back-end of the central dogma, have become increasingly important in recent years [1]. Modifications occurring on proteins exert their effects by influencing the structure, conformation, or stability of proteins, thereby regulating the growth and development of the organism, its autoimmunity, and its response to the environment [1,2,3].

Acylation refers to the process in which a series of short-chain acyl-coenzyme A binds to the ε-amino group of protein lysine for modification, with acetylation being the most typical [4]. These acyl-coenzymes are converted from corresponding short-chain fatty acids (SCFAs) under the action of Acyl-CoA Synthetase Short Chain Family Member 2 (ACSS2) and participate in the enzymatic modification of histones or non-histones [5]. Supplementing corresponding SCFAs can increase the relative level of the corresponding acylation. Due to the similar domains of acylations, there is a series of identical regulatory enzymes. The relative levels of different acylations within cells depend on the competition of different acyl-coenzymes for the “writer”. As an important part of PTMs, acylation is a bridge connecting metabolism and gene expression [6]. In the past decade of research, with the improvement in mass spectrometry sensitivity, researchers have identified propionylation, butyrylation, crotonylation, malonylation, succinylation, glutarylation, 2-hydroxyisobutyrylation, β-hydroxybutyrylation, lactylation, benzoylation, and acetoacetylation [7,8,9,10,11,12,13,14,15,16]. Although acylation is not as well-known as acetylation, it exhibits broad application prospects in various fields due to the diversity and fineness of its regulation.

The organism relies on a complex and elaborate energy metabolism network to meet the energy demands of life activities under different conditions. During starvation, fasting, or long-term exercise, the organism mobilizes stored fatty acids to supply energy through ketone body metabolism (Figure 1A) [17]. Fatty acids in adipose tissue are oxidized to generate ketone bodies, which are transported to various tissues through the blood and enter the tricarboxylic acid cycle to produce energy. Ketone bodies include hydroxybutyric acid, acetoacetic acid, and acetone, with most being β-hydroxybutyric acid (BHB). In addition to serving as an energy source, BHB can also improve the aging condition and can be used as an antioxidant and a target for disease treatment [18,19,20].

In the early stage, the mechanism of action of BHB on epigenetics was mainly considered to be the promotion of acetylation modification by inhibiting histone deacetylases (HDACs) [21]. Moreover, BHB can covalently modify histone lysine. This modification process is achieved through the activation of BHB into the thioester form of BHB-CoA, and the hydroxyl group bound to the lysine residue enhances its binding ability to hydrogen bonds [13,22]. This modification changes the structure of histones and their packaging of internal DNA, thereby regulating gene transcription. The level of histone lysine β-hydroxybutyrylation (Kbhb) depends on the level of BHB. Under starvation conditions, the level of BHB in the blood increases, promoting a 3.57–39.05-fold increase in histone Kbhb [13]. This change activates the expression of genes related to amino acid catabolism (alanine, aspartate, and glutamate metabolism), circadian rhythm, redox balance (selenoamino acid metabolism; cysteine and methionine metabolism), the peroxisome proliferator-activated receptor (PPAR) signaling pathway, and oxidative phosphorylation. In addition to regulating gene expression by modifying histones, Kbhb can also occur on non-histones. Non-histone Kbhb can affect the conformation of proteins, thereby changing the activity of enzymes, and can also reduce the phosphorylation and acetylation levels, thus affecting their functions [23,24,25].

Since Kbhb has only been known for a relatively short time, the current understanding of it is still insufficient. Currently, the presence of Kbhb has only been reported in humans, mice, and Ustilaginoidea virens, while research on Kbhb in pigs is still blank. The research on β-hydroxybutyric acid in pigs is mainly related to obesity. The BHB metabolic levels in obese pigs and lean meat pigs are not the same. Fatty pigs have a lower BHB content [26]. Lu et al. used pigs and mice as models for obesity research and found that BHB in a ketogenic diet can effectively manage the weight of obesity models [27]. As the most important meat-producing animal for humans, it is of great significance to further investigate the impact of Kbhb on obesity and its potential for fattening.

Given that the liver is the site of ketone body synthesis and plays a crucial role in the metabolism of BHB, we have conducted this study to investigate the level of Kbhb in pig liver, in order to analyze the specific modified proteins and their functions, thereby understanding the potential role of Kbhb in liver metabolism.

## 2. Materials and Methods

### 2.1. Experimental Design

The animal experimental protocol for this study was approved by the Animal Ethics Committee of Anhui Agricultural University under permit no. AHAU20201025. For the piglet experiment, one healthy male three-way hybrid piglet [(Duroc × Landrace) × Large White] (5 weeks of age) purchased from Lu ‘an Shitun Husbandry was carefully selected. The animal was anesthetized via intramuscular injection of ketamine (20 mg/kg) combined with diazepam (0.5 mg/kg) and atropine (0.05 mg/kg). After ventral positioning, the abdominal surface was sterilized with 75% ethanol. A midline laparotomy incision was performed to expose the hepatic lobes under sterile conditions. Approximately 5 g of liver tissue (left lateral lobe) was excised using disposable scalpel blades, immediately weighed, flash-frozen in liquid nitrogen to prevent protein degradation, and stored at −80 °C for subsequent analysis.

### 2.2. Cell Culture

The porcine liver cell line PICM-19 was purchased from bluefcell. Briefly, cells were cultured in DMEM medium supplemented with 10% fetal bovine serum and 1% penicillin and streptomycin antibiotic mixture (Invitrogen, Carlsbad, CA, USA). This cultivation was carried out at a temperature of 37 °C in a humidified atmosphere containing 5% CO_2_ and 95% air. BHB (10 mM, MCE; Cat. No. HY-113378) was dissolved in DMSO or water as per the manufacturer’s instructions [23].

### 2.3. Protein Extraction and Western Blot

The cell sample was ground with liquid nitrogen into powder and transferred to a 5 mL centrifuge tube. Four volumes of lysis buffer (1% Sodium dodecyl sulfate (SDS), 1% protease inhibitor cocktail (III-VII, Merck Millipore, Darmstadt, Germany)) were added to the cell powder, followed by sonication for three minutes on ice using a high-intensity ultrasonic processor (Scientz, Ningbo, China). For PTM experiments, inhibitors were also added to the lysis buffer, e.g., 3 μM Trichostatin A (TSA) and 50 mM Nicotinamide (NAM) for acetylation, and 1% phosphatase inhibitor (B15002, Selleck, Houston, TX, USA) for phosphorylation. The remaining debris was removed by centrifugation at 12,000× *g* at 4 °C for 10 min. Finally, the supernatant was collected, and the protein concentration was determined using a BCA kit (P0012, Beyotime, Shanghai, China) according to the manufacturer’s instructions.

Porcine liver and cells were collected, centrifuged, and lysed in RIPA buffer (Servicebio, Wuhan, China) for Western blotting. The collected samples were boiled for 5 min and electrophoresed on a 12.5% SDS—PAGE gel. The nitrocellulose membrane was blocked with 5% milk and incubated overnight with antibodies against pan-Kbhb (1:1000, PTM-1201, PTMBIO, Hangzhou, China), tubulin (1:1000, PTM-6414, PTMBIO, Hangzhou, China), and HMGCS2 (1:200, ab137043, Abcam, Cambridge, UK). After washing, the membranes were incubated with an HRP-labeled secondary antibody (Sangon, D111042-0100, Shanghai, China) for 2 h at room temperature and then photographed with chemiluminescent solution.

### 2.4. Trypsin Digestion

The protein sample was added to 1 volume of pre-cooled acetone, vortexed to mix, added to 4 volumes of pre-cooled acetone, and precipitated at −20 °C for 2 h. The precipitate was washed 2–3 times with pre-cooled acetone. Protein samples were dissolved in 200 mM Tetraethylammonium bromide (TEAB, Melbourne, Australia) and ultrasonically dispersed. Sequencing Grade Modified Trypsin (Promega, V5111, Madison, WI, USA) was added overnight at a 1:50 trypsin-to-protein mass ratio for the first digestion step. The sample was reduced with 5 mM dithiothreitol for 30 min at 56 °C and alkylated with 11 mM iodoacetamide for 15 min at room temperature in darkness. Finally, the peptides were desalted on a Strata X SPE column.

### 2.5. Affinity Enrichment of Kbhb Peptides

To enrich the modified peptides, tryptic peptides dissolved in Nonidet P-40, EDTA, Tris-HCl, NaCl (NETN) buffer (100 mM NaCl, 1 mM EDTA, 50 mM Tris-HCl, 0.5% NP-40, pH 8.0) were incubated with prewashed Anti-β-hydroxybutyryl Lysine Antibody Conjugated Agarose Beads (PTM-1204, PTM-Bio, Hangzhou, China) at 4 °C overnight with gentle shaking. The beads were then washed four times with NETN buffer and twice with H_2_O. The bound peptides were eluted from the beads using 0.1% trifluoroacetic acid. Finally, the eluted fractions were combined and vacuum dried. For LC-MS/MS analysis, the resulting peptides were desalted using C18 ZipTips (Millipore, Darmstadt, Germany) according to the manufacturer’s instructions.

### 2.6. LC-MS/MS Analysis

The tryptic peptides were dissolved in solvent A and directly loaded onto a homemade reversed-phase analytical column (15 cm length, 100 μm i.d., packed with ReproSil-Pur C18-AQ 1.9 μm resin, 120 Å pore size; Dr. Maisch, Ammerbuch, Germany #r119.aq). A Vanquish Neo UPLC system (ThermoFisher Scientific, Waltham, MA, USA) with trap column (ThermoFisher Scientific, Waltham, MA, USA, PEPMAP NEO C18 5 um 300 um × 5 mm trap 3PK 1500 Bar, #174500, Waltham, MA, USA) was used. The mobile phase consisted of solvent A (0.1% formic acid in water) and solvent B (0.1% formic acid, 80% acetonitrile/in water). Peptides were separated with the following gradient: 0–0.5 min, 4%B; 0.5–0.6 min, 4–8%B; 0.6–13.6 min, 8–22.5%B; 13.6–20.5 min, 22.5–35%B; 20.5–20.9 min, 35–55%B; 20.9–21.4 min. 55–99%B; 21.4–22.6 min, 99%B, at a constant flow rate of 400 nL/min on a Vanquish Neo UPLC system (ThermoFisher Scientific, Waltham, MA, USA).

The separated peptides were analyzed using an Orbitrap Astral instrument operated in data-independent acquisition (DIA) mode with a nano-electrospray ion source. The electrospray voltage applied was 1900 V. The DIA parameters were configured as follows: Precursor scan range: 380–980 *m*/*z* at 240,000 resolutions. DIA windows: 24 variable windows (400–416, 416–438, 438–466, 466–496, 496–528, 528–562, 562–598, 598–636, 636–676, 676–718, 718–762, 762–808, 808–856, 856–906, 906–946, 946–980 *m*/*z*). DIA window width: 2 *m*/*z*. Window overlap: 0 *m*/*z*. MS/MS scan range: 150–2000 *m*/*z* at 80,000 resolution (first mass fixed at 150.0 *m*/*z*). HCD fragmentation was performed at 25% normalized collision energy. The automatic gain control target was set to 500% with a maximum injection time of 3 ms.

### 2.7. Database Search

The resulting MS/MS data were processed using SpectronAut (v.18). Tandem mass spectra were obtained for Sus_scrofa_9823_PR_20240407. fasta (46,176 entries, Uniprot) and concatenated with the reverse-decoy database. Trypsin/P was used as the cleavage enzyme, allowing for up to four missing cleavages. The mass tolerance for the precursor ions was set to 20 ppm in the first search and 5 ppm in the main search. The mass tolerance for the fragment ions was set to 0.02 Da. Carbamidomethylation on Cys was specified as a fixed modification, and oxidation on methionine, β-hydroxybutyrylation on lysine, and acetylation on the protein N-terminus were specified as variable modifications. The FDR for protein, peptide and PSM identification in Spectronaut software was set at 1%. To obtain high-quality results, the results of the database analysis were further filtered, identifying proteins containing at least one peptide segment.

### 2.8. Functional Annotation

We identified the modified proteins and used the EggNOG-mapper software (v2.1.12) to extract the GO ID for each protein based on the EggNOG database. The proteins were functionally annotated by categorizing them according to their cellular components, molecular functions, and biological processes. We annotated protein pathways based on the KEGG pathway database and used the BLAST (2.15.0, blastp, e value ≤ 1 × 10^−4^) tool to compare the identified modified proteins. For the BLAST comparison results of each sequence, we selected the highest-scoring match for annotation. WolF Psort software (v3.0) was used to annotate the subcellular localization of the proteins. The MoMo analysis tool based on the Motif-X algorithm was used to analyze the motif features of the modified residues [28].

### 2.9. Targeted Metabolome

The cell pellet was resuspended in 100 μL of ultrapure water (Milli-Q^®^ IQ 7000, Merck Millipore, Darmstadt, Germany). Then, 50 μL of the suspension was mixed with 200 μL of LC-MS grade methanol (#34860-1L-R, Merck, Darmstadt, Germany, pre-cooled at −20 °C). The mixture was vortexed at 2500 r/min for 2 min. Subsequently, it was rapidly frozen in liquid nitrogen for 5 min, taken out and thawed on ice for 5 min, and then vortexed again for 2 min to ensure homogeneity. This cycle was repeated three times. After centrifugation at 12,000 r/min for 10 min at 4 °C, 200 μL of the supernatant was transferred to a new centrifuge tube. The tube was placed in a −20 °C refrigerator for 30 min and then centrifuged again at 12,000 r/min for 10 min at 4 °C.

After passing 180 μL of the supernatant through a protein precipitation plate (#96CD2025-Q-FX, Agela, Tianjin, China), it was analyzed by Ultra Performance Liquid Chromatography (UPLC) (Waters ACQUITY H—Class D, Milford, CT, USA) and Tandem Mass Spectrometry (MS/MS) (QTRAP^®^ 6500+).

Liquid Chromatography Conditions: Column: ACQUITY UPLC^®^ BEH Amide (1.7 μm, 2.1 × 100 mm; Waters #186004801, Milford, CT, USA). Mobile Phase: A: ultrapure water (10 mM ammonium acetate, 0.3% ammonia water). B: 90% acetonitrile/water (*v/v*). Gradient: 0 ⟶ 1.2 min, 5:95 A/B (*v/v*); 8 min, 30:70 A/B (*v/v*); 9.0 ⟶ 11 min, 50:50 A/B (*v/v*); 11.1 ⟶ 15 min, 5:95 A/B (*v/v*). Flow Rate: 0.4 mL/min. Column Temperature: 40 °C. Injection Volume: 2 μL.

Mass Spectrometry Conditions: Ionization: ESI with positive/negative polarity switching. Ion Spray Voltage: +5500 V (POS), −4500 V (NEG). Source Temperature: 500 °C. Gas Flow Rates: Curtain Gas 35 psi.

Quantitative Analysis: Standard Solutions: Calibration curves (NADPH and D-Glucose: 0.1–150,000 ng/mL, others: 0.01–15,000 ng/mL) prepared using MS-certified metabolite standards (IROA Technologies #MSML-LEO, Amsterdam, The Netherlands). All standard substances were purchased from Sigma-Aldrich with a purity greater than 99%. Data Processing: raw data were integrated in MultiQuant™ 3.0.3 (Sciex, Washington, DC, USA) with default setting.

For the remaining 50 μL of the cell suspension, it was subjected to three cycles of repeated freezing in liquid nitrogen and thawing. After centrifugation at 12,000 r/min for 10 min, the supernatant was collected, and the protein concentration was determined by the BCA method. The metabolomic data of the test samples were corrected based on the protein concentrations of the parallel samples.

## 3. Results

### 3.1. Global Profiling of Kbhb in Porcine Liver

To confirm the presence of β-hydroxybutyrylation (Kbhb) in porcine liver, we conducted Western blotting using a Pan-Kbhb antibody. Distinct bands corresponding to Kbhb proteins were observed across the entire proteome, indicating widespread distribution of this modification (Figure 1B). To further characterize the Kbhb substrates, we performed high-resolution mass spectrometry (MS) analysis using the advanced Orbitrap Astral platform. This approach identified 4982 Kbhb sites on 2122 proteins in porcine liver tissues (Figure 1C, Appendix A), with an average of 2.35 modification sites per protein under a stringent false discovery rate (FDR) threshold of <1%. Notably, our dataset represents the largest β-hydroxybutyrylome reported to date, surpassing previous studies in humans, mice, and other species (Figure 1D) [23,24,29,30,31,32,33,34,35].

The majority of modified peptides (7–20 amino acids in length) aligned with optimal tryptic digestion and enrichment efficiency, validating our experimental workflow (Appendix A). Among the 2122 modified proteins, 1961 proteins (92.4%) harbored fewer than five Kbhb sites, while 26 proteins exhibited >10 modification sites, suggesting potential functional hotspots (Appendix A). The lysine modification ratio (modified lysines/total lysines per protein) was <10% for 1428 proteins (67.3%), with eight proteins displaying ratios >0.5. The top three proteins with the highest modification ratios included Signal sequence receptor subunit 2 (A0A4X1VZX8), Small ribosomal subunit protein uS14 (A0A287AQ67), and Hemoglobin subunit alpha (P01965) (Appendix A), implicating roles in protein translocation, translation, and oxygen transport.

Subcellular localization prediction via WolF Psort revealed that Kbhb-modified proteins were predominantly localized in the cytoplasm (808 proteins, 38.08%), followed by mitochondria (384 proteins, 18.1%), nucleus (325 proteins, 15.32%), extracellular matrix (220 proteins, 10.37%), plasma membrane (141 proteins, 6.64%), cytoplasm/nucleus dual localization (98 proteins, 4.62%), endoplasmic reticulum (98 proteins, 4.62%), and other compartments (48 proteins, 2.26%) (Figure 1E). This distribution aligns with the metabolic versatility of liver cells and the mitochondrial origin of BHB biosynthesis.

### 3.2. Motif Analysis of Kbhb Sites

The amino acid environment surrounding modification sites plays a critical role in determining enzyme-substrate specificity, with these residue patterns referred to as motifs. To uncover the underlying principles of enzyme–substrate interactions, we analyzed the sequence context of Kbhb-modified lysines. By examining the distribution of amino acids within ±10 residues of modified lysines, we identified three conserved residues—leucine (L), phenylalanine (F), and valine (V)—at the +1 position relative to the modified lysine (Figure 2A, Appendix A). Motif-X analysis confirmed these motifs as statistically significant signatures of Kbhb, with the highest enrichment scores (Figure 2B). Consistent with this, Icelogo heatmaps revealed elevated frequencies of L/F/V residues immediately downstream of modified lysines (Figure 2C), suggesting that these hydrophobic residues may facilitate enzyme recognition or stabilize the modified lysine microenvironment.

These findings provide valuable insights into the sequence preferences of Kbhb-modifying enzymes and highlight potential mechanisms underlying substrate specificity. The conserved presence of L/F/V residues at the +1 position may reflect their role in enhancing enzyme binding affinity or influencing the structural dynamics of the modified lysine. These motifs could serve as predictive markers for identifying novel Kbhb substrates and further elucidate the regulatory networks governed by this modification.

### 3.3. Functional Enrichment of Kbhb Proteins

To elucidate the biological roles of Kbhb in porcine liver, we performed multi-omics functional annotation. COG/KOG classification revealed enrichment in processes such as post-translational modification, signal transduction, translation, and lipid metabolism (Figure 3A). Gene Ontology (GO) analysis further highlighted Kbhb involvement in fatty acid β-oxidation, oxidative phosphorylation, amino acid metabolism, and energy metabolism (Appendix A). KEGG pathway mapping identified significant enrichment in carbohydrate metabolism, TCA cycle, glycolysis/gluconeogenesis, fatty acid degradation, and cytochrome P450-mediated xenobiotic metabolism (Figure 3B). These findings underscore Kbhb’s regulatory potential in hepatic energy homeostasis and detoxification.

### 3.4. Histone Kbhb in Porcine Liver

Histone Kbhb, widely found in model animals such as humans and mice and recently found in fungi [13,34], was also identified in porcine liver. A total of 16 Kbhb sites were found to be located on histone proteins, including H1 (ten sites), H2A (one site), H2B (two sites), and H3 (three sites) (Figure 4, Appendix A). A conservative analysis found that different species H1K85bhb, H1K90BHB, H1K106BHB, H2BK5BHB, H3K23BHB, and H3K56BHB are highly conserved between different mammals, maybe as a genetic marker of chromatin in evolution. Some sites such as H2AK95BHB and H3K122BHB were consistent in humans and pigs, and there was no Kbhb site that remained consistent only between mice and pigs. Among the six sites conserved in mammals, H2BK5BHB, H3K23BHB, and H3K56BHB have been reported to promote gene expression, and H3K56BHB is known to link to chromatin super enhancers. The widespread presence of histone Kbhb in porcine liver suggests the potential existence of an epigenetic transcriptional regulatory mechanism that warrants further exploration.

### 3.5. Functional Insights into Kbhb-Regulated Energy Metabolism

Liver, as the central metabolic hub in animals, plays a pivotal role in lipid synthesis and catabolism. Fatty acids, primarily metabolized in the liver, undergo β-oxidation and ketogenesis to generate energy. These pathways are critical for energy production, particularly under conditions of glucose scarcity, where acetyl-CoA derived from fatty acid breakdown is channeled into the tricarboxylic acid (TCA) cycle to fuel ATP generation. Our KEGG pathway enrichment analysis revealed that the majority of proteins involved in fatty acid synthesis, β-oxidation, and the TCA cycle are modified by Kbhb, suggesting that Kbhb may regulate these pathways by altering protein structure or function (Figure 5). We selected the protein HMGCS2 involved in ketone body metabolism for WB verification. The results indicated that the addition of BHB enhanced the Kbhb level of HMGCS2 (Appendix A). Notably, similar enrichment patterns for the TCA cycle, fatty acid degradation, and glycolysis pathways were observed in Kbhb-modified proteomes of mice and the rice-pathogenic fungus Ustilaginoidea virens [23,34], highlighting the evolutionary conservation of Kbhb’s regulatory role in energy metabolism. To further explore the mechanistic basis of this regulation, we conducted domain-specific enrichment analysis of Kbhb sites. Intriguingly, Kbhb modifications were predominantly localized to functional domains of key enzymes involved in fatty acid degradation, including Propionyl-CoA carboxylase domain (essential for propionate metabolism and odd-chain fatty acid oxidation), Thiolase C-terminal and N-terminal domains (critical for thiolytic cleavage in β-oxidation), and Acyl-CoA dehydrogenase N-terminal and C-terminal domains (key to fatty acid dehydrogenation) (Appendix A). These findings suggest that Kbhb modifications within these catalytic or substrate-binding regions may directly influence enzyme activity or substrate affinity. For example, lysine residues in the Thiolase C-terminal domain are involved in CoA-binding and dimerization—processes that could be modulated by β-hydroxybutyrylation. Similarly, modifications in the Acyl-CoA dehydrogenase domains might alter FAD cofactor interactions, thereby impacting electron transfer efficiency during β-oxidation. Given the mitochondrial localization of these enzymes (Figure 1E) and the organelle’s central role in energy metabolism, we hypothesize that Kbhb serves as a mitochondrial retrograde signal, fine-tuning metabolic flux in response to nutrient availability.

### 3.6. Kbhb Regulates Hepatic Energy Metabolism

To investigate the impact of β-hydroxybutyrylation (Kbhb) on fatty acid oxidation and energy supply, we treated cells with BHB, the substrate for Kbhb, to elevate the Kbhb levels. Western blotting confirmed that 10 mM BHB treatment significantly increased global Kbhb levels in hepatocytes (Figure 6A). Subsequently, we employed LC-MS/MS-based targeted metabolomics to quantify intracellular metabolites related to energy metabolism. Principal component analysis (PCA) demonstrated excellent reproducibility across replicates (Figure 6B). A total of 70 metabolites were detected, and using a threshold of |fold change| > 1.2 and *p* < 0.05, we identified eight upregulated and nine downregulated metabolites following BHB treatment (Figure 6C, Appendix A).

Functional enrichment analysis of these differentially regulated metabolites revealed significant involvement in glyoxylate and dicarboxylate metabolism, butanoate metabolism, the TCA cycle, and pyruvate metabolism (Figure 6C). Notably, the TCA cycle exhibited the highest number of altered metabolites, with six key intermediates showing significant changes (Appendix A). Among these, α-ketoglutarate, pyruvate, malate, succinate, and fumarate were upregulated, while acetyl-CoA was downregulated. Strikingly, α-ketoglutarate displayed a 9.56-fold increase, indicating a robust metabolic response to BHB treatment (Appendix A). These findings suggest that Kbhb may drive metabolic reprogramming by modulating the activity of key metabolic enzymes. The upregulation of TCA cycle intermediates, particularly α-ketoglutarate, points to enhanced flux through this pathway, potentially redirecting carbon sources toward energy production under conditions of elevated Kbhb. The observed decrease in acetyl-CoA levels may reflect its increased utilization in the TCA cycle or its conversion to other metabolites. Collectively, these results highlight the role of Kbhb as a metabolic regulator, linking post-translational modifications to cellular energy homeostasis. Further studies are needed to elucidate the specific mechanisms by which Kbhb influences enzyme activity and metabolic flux.

## 4. Discussion

The liver, as the central metabolic organ in animals, orchestrates intricate pathways to regulate metabolite synthesis. Beyond their roles in energy production, certain metabolites—such as acetyl-CoA, succinyl-CoA, and lactate—act as epigenetic factors by modifying proteins. Ketone body metabolism serves as a backup energy source during glucose scarcity. The liver generates ketone bodies via fatty acid oxidation, which are transported to peripheral tissues via blood and ultimately converted into acetyl-CoA for ATP production through the TCA cycle. Ketone bodies primarily include β-hydroxybutyrate (BHB), acetoacetate, and acetone, with BHB constituting over 70% of total ketones. In humans and mice, both BHB and acetoacetate have been reported to function as epigenetic regulators by modifying histone lysine residues to modulate gene expression [13,16]. Among these, BHB-dependent Kbhb has been extensively studied and implicated in diverse biological processes, including DNA damage repair, drug addiction, immune remodeling, energy metabolism, and lipid metabolism [23,24,31,35]. Recently, Chen et al. identified widespread Kbhb in Ustilaginoidea virens, a rice-pathogenic fungus, where it regulates fungal virulence [33]. However, research on Kbhb in livestock species, particularly pigs—a critical agricultural animal and valuable model for liver disease studies—remains unexplored [36]. Given the evolutionary conservation of Kbhb, we hypothesized its presence and metabolic regulatory roles in pigs. Here, we systematically characterized the porcine hepatic β-hydroxybutyrylome to identify Kbhb substrates and their functional implications, and we confirmed that BHB serves as the donor for Kbhb in porcine liver.

To comprehensively map Kbhb substrates, we employed the state-of-the-art Astral mass spectrometer for global profiling of porcine liver proteins. A total of 4982 Kbhb sites across 2122 proteins were identified, representing the largest dataset reported to date, underscoring the ubiquity of Kbhb in porcine liver. The biosynthetic pathway of BHB-CoA remains incompletely understood. One proposed route involves cytosolic/nuclear ACSS2-mediated conversion of BHB to BHB-CoA [36], followed by lysine modification via ketone body transferases (KBTs). Given that ACSS2 has recently been reported to catalyze the synthesis of lactyl-CoA from lactate, there may be substrate concentration-dependent competition between BHB and lactate for ACSS2 activity, with the dominant metabolic pathway (glycolysis or β-oxidation) determining the primary product [37,38]. Alternatively, mitochondrial β-oxidation may generate BHB-CoA, which non-enzymatically modifies lysine residues under the high-concentration microenvironment of the mitochondrial matrix [39]. However, research on this non-enzymatic mechanism remains limited, with P300-catalyzed Kbhb being the predominant pathway [31]. Subcellular localization revealed enrichment of Kbhb-modified proteins in the cytoplasm, mitochondria, and nucleus—consistent with BHB-CoA synthesis sites. While the porcine hepatic Kbhb protein distribution aligned with human and murine patterns, tissue-specific differences emerged: nuclear enrichment in human HEK293 cells versus mitochondrial dominance in mouse liver, suggesting species- and tissue-dependent metabolic adaptations [23,31].

After understanding the regulatory mechanism of Kbhb, it is crucial to clarify its substrate for the specific functional research. Kbhb occurs on both histones and non-histone proteins. The hydroxyl group introduced at lysine residues enhances the hydrogen bonding capacity, potentially recruiting chromatin-active factors to promote transcription [22]. Histone Kbhb has been linked to tumor metabolism and ferroptosis regulation [40,41]. In our dataset, 99.68% of Kbhb sites localized to non-histone proteins, highlighting functional diversity. Strikingly, proteins undergoing Kbhb modification that are involved in fatty acid β-oxidation were significantly enriched. Moreover, all five core enzymes in the fatty acid metabolism pathway—including ACSL1, ACADs, ECHS1, HADH, and ACAA2—were found to be Kbhb-modified. This finding extends the “localized precursor accumulation” hypothesis proposed by Ringel et al., by demonstrating that elevated intramitochondrial acyl-CoA pools generated during β-oxidation may directly serve as substrates for lysine crotonyltransferases, such as CBP/p300, thereby facilitating spatially coordinated enzymatic modification [39]. Domain enrichment analysis further identified Kbhb hotspots within functional domains of β-oxidation and ketogenesis proteins (e.g., Propionyl-CoA carboxylase, Thiolase, Acyl-CoA dehydrogenase), suggesting modulation of enzymatic activity. For instance, Koronowski et al. demonstrated that K405bhb in the C-terminal domain of AHCY suppresses enzyme activity, which was restored upon lysine-to-arginine mutation [23]. Qin et al. discovered that the ketogenic diet inhibits glycolysis and tumor cell proliferation by enhancing ALDOB-K108bhb to suppress its enzymatic activity and weaken its binding with fructose-1,6-bisphosphate [42]. Aging also increases the level of STAT1-K592bhb, inhibits the interaction between STAT1 and type-I interferon, and leads to a decline in antiviral activity. Inhibiting the level of STAT1kbhb can effectively improve this phenomenon [43]. In addition to regulating enzyme activity and protein–protein interactions, Kbhb also affects protein stability. The Kbhb of the transcription factor Snail can prevent the degradation of Snail by blocking the recognition of the E3 ubiquitin ligase FBXL14 [44].

Given the influence of Kbhb on a series of metabolic enzymes, it is necessary to understand the role of Kbhb in metabolic reprogramming in porcine liver cells. Synthesized in mitochondria, BHB is exported via SLC16A6 and distributed to peripheral tissues via monocarboxylate transporters (MCT1/2) [17]. In extrahepatic tissues, BHB is reconverted to acetyl-CoA for TCA cycle entry, explaining the widespread mitochondrial Kbhb observed here. While mitochondrial proteins are classically associated with acetylation and succinylation [45], BHB’s structural similarity to butyrate enables HDAC inhibition, potentially amplifying acetylation levels [17]. Paradoxically, BHB consumption via conversion to acetoacetate depletes succinyl-CoA, reducing succinylation—a phenomenon corroborated by our metabolomics data showing BHB-induced succinate accumulation. Although α-ketoglutarate (α-KG) increased by 9.56-fold, we did not observe a corresponding rise in succinyl-CoA, and succinate only increased by 2.16-fold, likely due to the substantial consumption of succinyl-CoA. Moreover, key TCA cycle intermediates, such as α-KG, malate, and fumarate, also exhibited varying degrees of elevation. We speculate that this may result from both increased succinate substrate levels and Kbhb-mediated modulation of TCA cycle proteins. However, given the complexity of metabolic regulation, the precise mechanisms underlying these effects require further investigation. Future studies should focus on elucidating the molecular mechanisms by which Kbhb modulates enzyme activity and metabolic pathways. For example, structural studies of Kbhb-modified enzymes could reveal how this modification influences substrate binding, catalytic efficiency, or protein–protein interactions. Additionally, the development of site-specific antibodies or chemical probes for Kbhb would enable more precise functional studies in vivo.

## 5. Conclusions

In summary, we present the first landscape of Kbhb in porcine liver, identifying 4982 sites on 2122 proteins, including 16 histone sites. Functional enrichment and experimental validation elucidated Kbhb’s role in energy metabolism and its dynamic regulatory mechanisms. This study expands the repertoire of Kbhb substrates in pigs and provides a foundation for exploring its biological significance in livestock physiology and disease. A notable limitation of this study lies in its preliminary exploration of Kbhb-associated proteins, without delving into specific biological contexts such as obesity or toxin-induced liver injury to investigate differential Kbhb patterns. Subsequent research will prioritize mechanistic elucidation of key proteins to further clarify the physiological and molecular functions of BHB.

## Figures and Tables

**Figure 1 biology-14-01183-f001:**
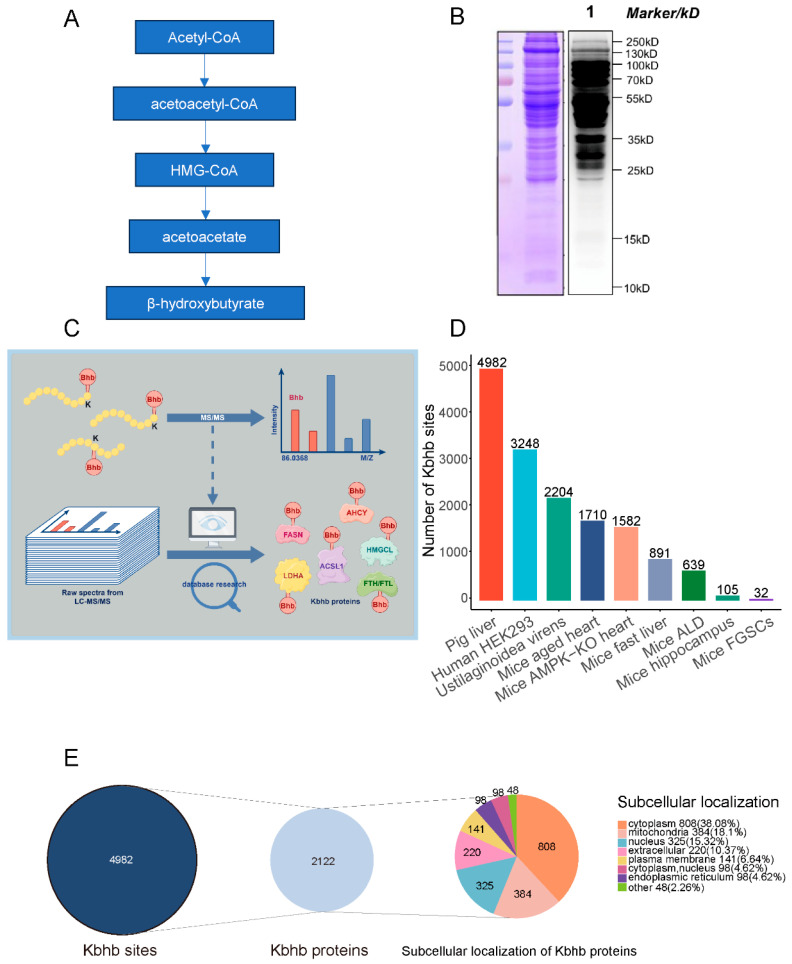
Global profiling of protein β-hydroxybutyrylome. (**A**) Metabolic process of ketone body. (**B**) Western blot of Kbhb level using pan-Kbhb antibody. (**C**) Mechanism diagram of this experiment. (**D**) Identified Kbhb sites from different spaces. (**E**) Statistical analysis and subcellular localization of Kbhb proteins.

**Figure 2 biology-14-01183-f002:**
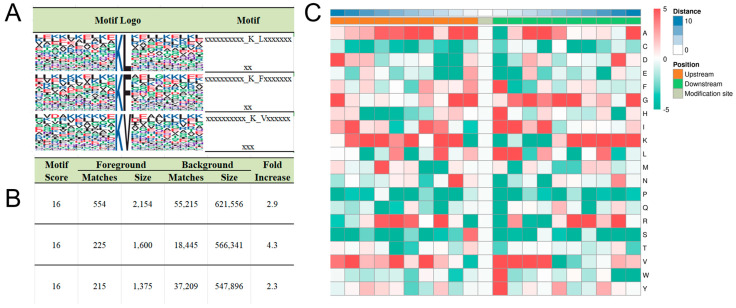
Motif analysis reveals conserved sequence patterns flanking Kbhb sites. (**A**) Top three ranked motif logo enriched by motif-X. (**B**) Score of top three enriched motif logos. (**C**) Heatmap showing the preference of the ten amino acids upstream and downstream of the modified K. Red means the score of amino acids is greater than 0, and green means scores less than 0.

**Figure 3 biology-14-01183-f003:**
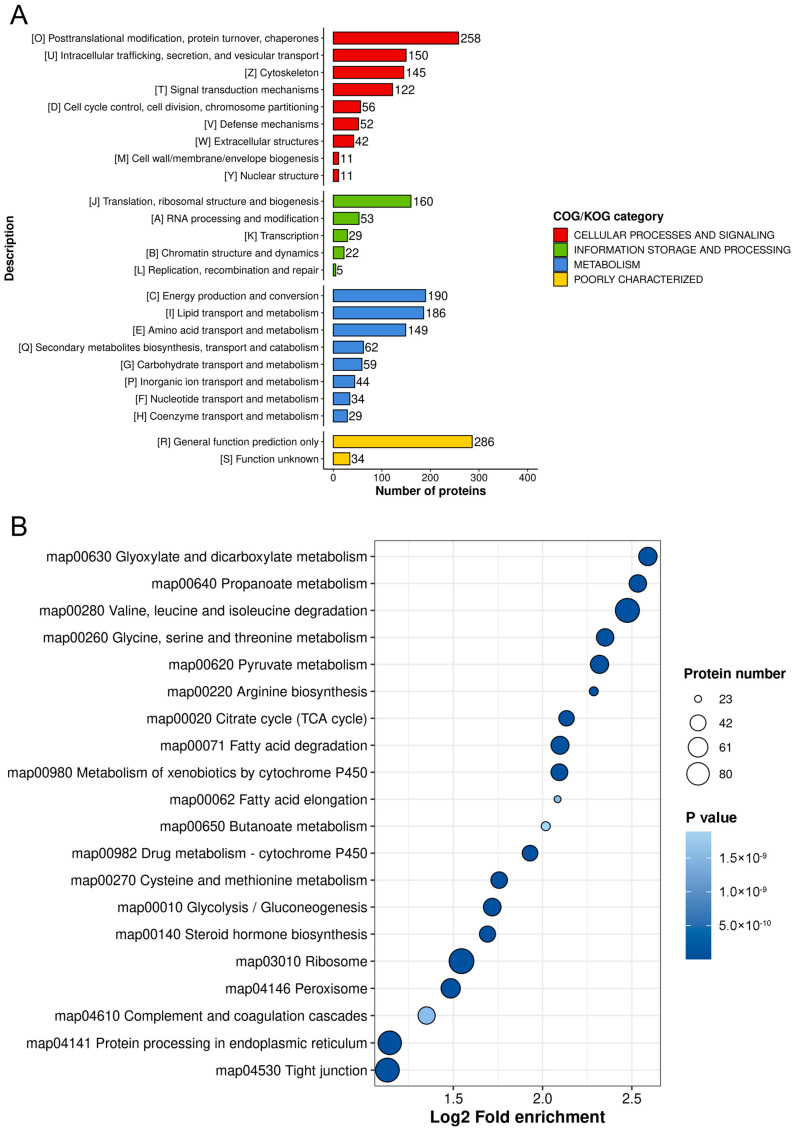
Functional enrichment of Kbhb proteins implicates key metabolic and regulatory pathways. (**A**) represents COG/KOG enrichment, and (**B**) represents KEGG enrichment of Kbhb proteins. The size of the plot represents the protein numbers enriched in the pathway. The letters in the brackets of the vertical coordinate represent the corresponding functional categories.

**Figure 4 biology-14-01183-f004:**
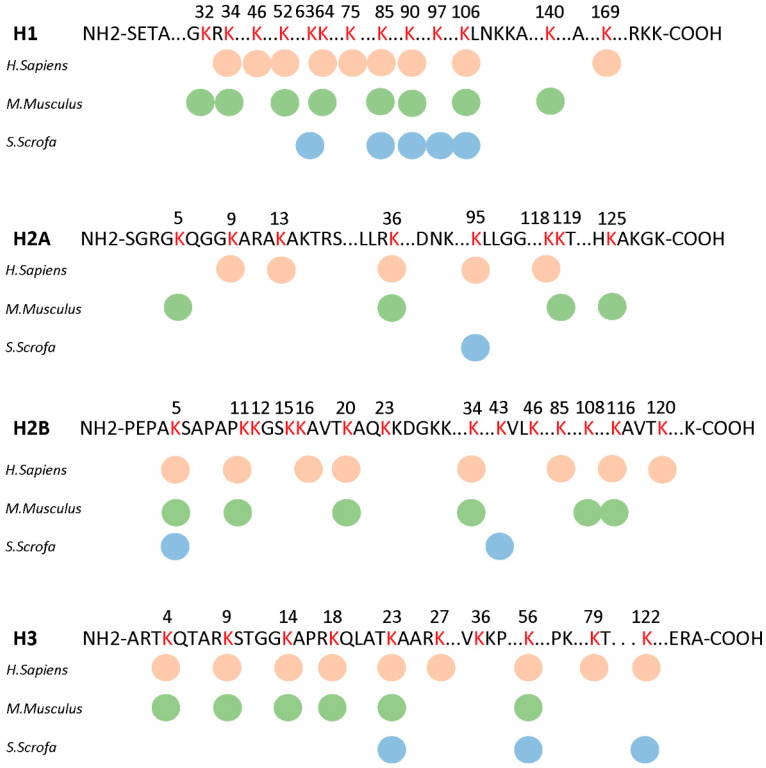
The distribution of Kbhb sites on histones. Conservation analysis of lysine sites modified by β-hydroxybutyrylation on histones in humans, mice, and pigs. Different colored dots represent different species. Modified lysines are marked in red.

**Figure 5 biology-14-01183-f005:**
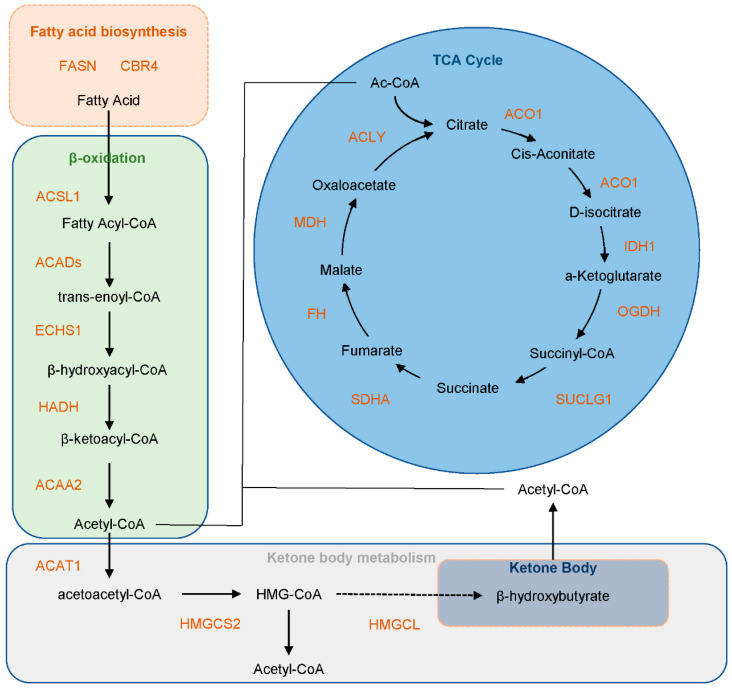
The situation of Kbhb protein in the core metabolic pathways. The figure shows the β-hydroxybutyrylation status of the main catalytic enzymes involved in fatty acid metabolism, β-oxidation, TCA cycle, and ketone body metabolism. The orange-colored proteins indicate that both expression and modification are simultaneously detected.

**Figure 6 biology-14-01183-f006:**
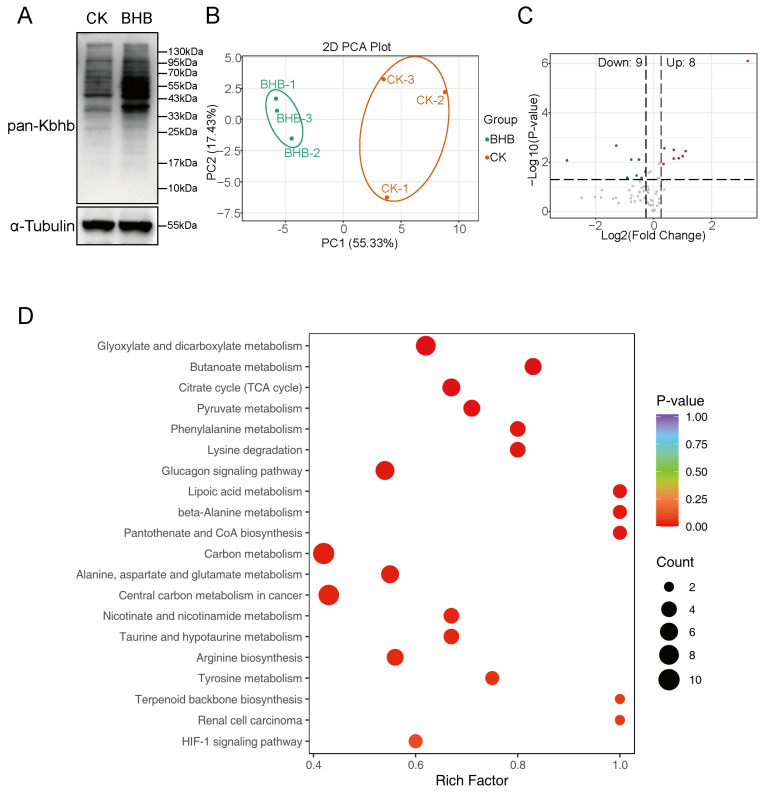
Metabolome revealing the metabolic reprogramming regulated by Kbhb. (**A**). WB showing the pan-Kbhb level before (CK) and after (BHB) 10 mM BHB treatment. (**B**) PCA analysis of samples using in metabolome. Red and green represents up- and downregulated metabolites. (**C**) Volcano showing different metabolites after BHB treatment. (**D**) KEGG enrichment of different metabolites. The size of the plot represents the numbers enriched in the pathway.

## Data Availability

The mass spectrometry proteomics data have been deposited in the ProteomeXchange Consortium (https://proteomecentral.proteomexchange.org) via the iProX partner repository with the dataset identifier IPX0011475000 (URL: https://www.iprox.cn/page/DSV021.html;?url=17552389629927TOL, passwords: zElY) with access date on 15 August 2025. The raw data of the metabolome were stored in the China National Center for Bioinformation (https://www.cncb.ac.cn/) with number PRJCA044753.

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
