# Peer review of "Global Profiling of Protein β-hydroxybutyrylome in Porcine Liver"

_biology, 2025, doi:10.3390/biology14091183_

Round 1

Reviewer 1 Report

Comments and Suggestions for Authors

The manuscript entitled, “Global profiling of protein β-hydroxybutyrylome in porcine liver” demonstrates a convincingly sound finding that unraveled Kbhb’s role in energy metabolism and various regulatory mechanisms. The effective profiling of porcine hepatic β-hydroxybutyrylome validates its application in livestock sector. The manuscript provides an elaborate and clear overview of the method adopted, whilst addressing the limitations of the former studies. A comprehensive statistical analysis that strengthens the validity of findings has been adopted. In addition, clear presentation of results through figures has been taken up. The current study findings are thus rational and can be of practical significance. In my opinion, the manuscript is authored well. But before the paper is considered for publication the authors must address the following points:

Abstract 1) Line 23 and 24 can be reconsidered as previous studies have reported pigs to be one among the ideal model for understanding metabolic diseases. A novel finding from the study can be added

2) Line 12-13 and wherever appropriate, consider revising the verb tense

3) Line 14 & 19, do not begin a sentence with abbreviation

4) Consider abbreviating wherever necessary

5) Consider including a “simple summary” section prior to abstract

Introduction

1) Line 78, revise the in-text cited reference numbers

2) The significance of studying Kbhb in pigs can be highlighted

3) The objective and the hypothesis of this study can be elaborated

4) In later section of introduction, summarise the practical significance of the work

Materials and method

1) Sub-section 2.1, the title can be revised as “Experimental design”

2) Line 89, revise the statement

3) Line 90-92, consider rephrasing these sentences and elaborate the sample collection process in about 50 words and include it as a separate sub-section

Results

1) Figure 2 & 3, a short description can be added

Discussion

1) Line 385-387- attribute the study findings, provide scientific explanation to improve understanding

2) Enhance the transitions while discussing each variables to improve clarity

3) Consider including a conclusion section highlighting study findings, significance, limitations of the study and future scope for further research. A stronger emphasis on the practical applications of findings would enhance impact

Author Response

Dear Reviewer:

   We are pleased to submit our revised manuscript entitled “Global profiling of protein β-hydroxybutyrylome in porcine liver” (Manuscript ID: biology-3789376) for publication consideration. ​We are extremely grateful for your time and patience in reading our manuscript, which helped us identify many minor issues. We carefully addressed all the issues which you raised. We carefully examined all the details you pointed out to ensure that nothing was missed. We believe that the revised manuscript has been significantly improved.

Our detailed point-by-point responses to each comment are as following:

The manuscript entitled, “Global profiling of protein β-hydroxybutyrylome in porcine liver” demonstrates a convincingly sound finding that unraveled Kbhb’s role in energy metabolism and various regulatory mechanisms. The effective profiling of porcine hepatic β-hydroxybutyrylome validates its application in livestock sector. The manuscript provides an elaborate and clear overview of the method adopted, whilst addressing the limitations of the former studies. A comprehensive statistical analysis that strengthens the validity of findings has been adopted. In addition, clear presentation of results through figures has been taken up. The current study findings are thus rational and can be of practical significance. In my opinion, the manuscript is authored well. But before the paper is considered for publication the authors must address the following points:

Abstract

1) Line 23 and 24 can be reconsidered as previous studies have reported pigs to be one among the ideal model for understanding metabolic diseases. A novel finding from the study can be added

Response: Thanks for your opinion. We had revised metabolic disease mechanisms to metabolic reprogramming. The revised sentence is as following: This study establishes pigs as a model for Kbhb research, linking it to energy regulation and providing insights into metabolic reprogramming. (L32-33)

2) Line 12-13 and wherever appropriate, consider revising the verb tense

Response: Thanks for your opinion. This is an excellent suggestion, and we have revised the verb tenses to the past tense. The modified sentence reads as follows: Here, we characterized the porcine hepatic β-hydroxybutyrylome using high-resolution mass spectrometry, identifying 4,982 Kbhb sites on 2,122 proteins—the largest dataset to date. (L21-23)

3) Line 14 & 19, do not begin a sentence with abbreviation

Response: Thanks for your opinion. We replaced the abbreviations at the beginning of the sentences with their full forms.

4) Consider abbreviating wherever necessary

Response: Thanks for your opinion. We have thoroughly revised the manuscript by adding abbreviations for frequently recurring technical terms, following the standard rule of defining abbreviations at first mention (full term followed by abbreviation in parentheses).

5) Consider including a “simple summary” section prior to abstract

Response: Thanks for your opinion. We have already added the "simple summary" section as required by the journal. (L11-16)

Introduction

  • Line 78, revise the in-text cited reference numbers

Response: Thanks for your opinion. We had revised the reference number you pointed out.

  • The significance of studying Kbhb in pigs can be highlighted

Response: Thanks for your opinion. We have added the significance of studying BHB and Kbhb on pigs, as follows: The research on β-hydroxybutyric acid in pigs is mainly related to obesity. The BHB metabolic levels in obese pigs and lean meat pigs are not the same. Fatty pigs have a lower BHB content. Lu et al. used pigs and mice as models for obesity research and found that BHB in a ketogenic diet can effectively manage the weight of obesity models. As the most important meat-producing animal for humans, it is of great significance to further investigate the impact of Kbhb on obesity and its potential for fattening. (L90-96) 

  • The objective and the hypothesis of this study can be elaborated

Response: Thanks for your opinion. We had rewritten the last paragraph of introduction to add the objective: Given that the liver is the site of ketone body synthesis and plays a crucial role in the metabolism of BHB, we have conducted this study to investigate the level of Kbhb in pig liver, in order to analyze the specific modified proteins and their functions, thereby understanding the potential role of Kbhb in liver metabolism. (L97-100)

  • In later section of introduction, summarise the practical significance of the work

Response: Thanks for your opinion. The response to this point is as the same as point 3.

Materials and method

  • Sub-section 2.1, the title can be revised as “Experimental design”

Response: Thanks for your opinion. We had revised it as your suggestion.

  • Line 89, revise the statement

Response: Thanks for your opinion. We had revised it as your suggestion.

  • Line 90-92, consider rephrasing these sentences and elaborate the sample collection process in about 50 words and include it as a separate sub-section

Response: Thanks for your opinion. We rewritten this part as your advice to make it clear. The animal was anesthetized via intramuscular injection of ketamine (20 mg/kg) combined with diazepam (0.5 mg/kg) and atropine (0.05 mg/kg). After ventral positioning, the abdominal surface was sterilized with 75% ethanol. A midline laparotomy incision was performed to expose the hepatic lobes under sterile conditions. Approximately 5g of liver tissue (left lateral lobe) was excised using disposable scalpel blades, immediately weighed, flash-frozen in liquid nitrogen to prevent protein degradation, and stored at -80°C for subsequent analysis.

Results

  • Figure 2 & 3, a short description can be added

Response: Thanks for your opinion. We have provided additional explanations for these two pictures as per your suggestion.

Discussion

  • Line 385-387- attribute the study findings, provide scientific explanation to improve understanding

Response: Thanks for your opinion. We had modified this part to: Strikingly, proteins undergoing Kbhb modification that are involved in fatty acid β-oxidation were significantly enriched. Moreover, all five core enzymes in the fatty acid metabolism pathway - including ACSL1, ACADs, ECHS1, HADH, and ACAA2 - were found to be Kbhb-modified. This finding extends the "localized precursor accumulation" hypothesis proposed by Ringel et al., by demonstrating that elevated intramitochondrial acyl-CoA pools generated during β-oxidation may directly serve as substrates for lysine crotonyltransferases, such as CBP/p300, thereby facilitating spatially coordinated enzymatic modification.

  • Enhance the transitions while discussing each variables to improve clarity

Response: Thanks for your opinion. We respectively added a transitional phrase at the beginning of these two paragraphs: “After understanding the regulatory mechanism of Kbhb, it is crucial to clarify its substrate for the specific functional research.” and “Given the influence of Kbhb on a series of metabolic enzymes, it is necessary to understand the role of Kbhb in metabolic reprogramming in porcine liver cells.”.(L430-431, L457-458)

3) Consider including a conclusion section highlighting study findings, significance, limitations of the study and future scope for further research. A stronger emphasis on the practical applications of findings would enhance impact

Response: Thanks for your opinion. This is a valuable advice. A comprehensive conclusion should encompass these components. We rewritten the conclusion as follow:

In summary, we present the first landscape of Kbhb in porcine liver, identifying 4,982 sites on 2,122 proteins, including 16 histone sites. Functional enrichment and ex-perimental validation elucidated Kbhb’s role in energy metabolism and its dynamic regulatory mechanisms. This study expands the repertoire of Kbhb substrates in pigs and provides a foundation for exploring its biological significance in livestock physiology and disease. A notable limitation of this study lies in its preliminary exploration of Kbhb-associated proteins, without delving into specific biological contexts such as obe-sity or toxin-induced liver injury to investigate differential Kbhb patterns. Subsequent research will prioritize mechanistic elucidation of key proteins to further clarify the physiological and molecular functions of BHB.

Reviewer 2 Report

Comments and Suggestions for Authors

The manuscript is well written with no major flaws. The findings linking Kbhb to energy regulation provide a solid foundation for future research into metabolic disease mechanisms. The integration of computational approaches with experimental validation to map modification sites is well-executed. However, a few clarifications and improvements would strengthen the manuscript further:

  • It is unclear whether the top candidate proteins identified through computational analyses were experimentally validated, for example via qPCR or Western blot. Including such validation would enhance the strength of the conclusions.
  • Fig. 1. The font size within the panels should be increased for better readability. Additionally, the y-axis in panel D needs a clear label indicating what the numerical values represent.
  • Fig. 2. The legend mentions that red indicates the amino acid score, but it is unclear what the green color represents. Please clarify all color coding in the legend.
  • The methods section would benefit from more detail regarding the Western blot procedure, including antibody concentrations, incubation conditions, and detection methods used.
  • Fig. 3. The alphabetical labels in brackets within the figure are not clearly explained. These should be defined in the legend for reader clarity.
  • Fig. 5. The arrow indicating β-oxidation needs adjustment for clarity. The legend states that orange-colored proteins reflect both expression and modification, but it would be more informative if the figure itself included directional arrows (e.g., up/down) to indicate regulation trends. Consider adding this information directly to the figure.
  • The future perspectives section must be elaborated.
  • The authors state that "Kbhb occurs on both histone and non-histone proteins," but the discussion is disproportionately focused on histones given the prediction and identification that cytoplasmic proteins are more in abundance. A more balanced analysis or an explanation for the limited insights into non-histone targets would be valuable.
  • Update the reference list to include the most recent and relevant literature.

Author Response

Dear Reviewer:

   We are pleased to submit our revised manuscript entitled “Global profiling of protein β-hydroxybutyrylome in porcine liver” (Manuscript ID: biology-3789376) for publication consideration. ​We are extremely grateful for your time and patience in reading our manuscript, which helped us identify many minor issues. We carefully addressed all the issues which you raised. We carefully examined all the details you pointed out to ensure that nothing was missed. We believe that the revised manuscript has been significantly improved.

Our detailed point-by-point responses to each comment are as following:

The manuscript is well written with no major flaws. The findings linking Kbhb to energy regulation provide a solid foundation for future research into metabolic disease mechanisms. The integration of computational approaches with experimental validation to map modification sites is well-executed. However, a few clarifications and improvements would strengthen the manuscript further:

The manuscript is well written with no major flaws. The findings linking Kbhb to energy regulation provide a solid foundation for future research into metabolic disease mechanisms. The integration of computational approaches with experimental validation to map modification sites is well-executed. However, a few clarifications and improvements would strengthen the manuscript further:

It is unclear whether the top candidate proteins identified through computational analyses were experimentally validated, for example via qPCR or Western blot. Including such validation would enhance the strength of the conclusions.

Response: Thanks for your opinion. This comment is of great help to the evidence chain of our paper. We detected the Kbhb level of the core enzyme HMGCS2 in ketone body metabolism through WB and found that the BHB level effectively increased its Kbhb level. The results are presented in the supplementary materials.

Fig. 1. The font size within the panels should be increased for better readability. Additionally, the y-axis in panel D needs a clear label indicating what the numerical values represent.

Response: Thanks for your opinion. In order to enhance the quality of the figure, we made modifications in accordance with the suggestions. The modified figure is presented below.

Fig. 2. The legend mentions that red indicates the amino acid score, but it is unclear what the green color represents. Please clarify all color coding in the legend.

Response: Thanks for your opinion. This comment has well supplemented our explanation of Figure 2. Both red and green represent amino acid scores, with red indicating values greater than 0 and green indicating values less than 0. We have made the corresponding modifications.

The methods section would benefit from more detail regarding the Western blot procedure, including antibody concentrations, incubation conditions, and detection methods used.

Response: Thanks for your opinion. The details of WB are crucial to the experiment. We have rewritten this part. The rewritten paragraph is as follows:

Porcine liver and cells were collected, centrifuged, and lysed in RIPA buffer (Ser-vicebio, China) for Western blotting. The collected samples were boiled for 5 minutes and electrophoresed on a 12.5% SDS‒PAGE gel. The nitrocellulose membrane was blocked with 5% milk and incubated overnight with antibodies against pan-Kbhb (1:1000, PTM-1201, PTMBIO), tubulin (1:1000, PTM-6414, PTMBIO) and HMGCS2 (1:200, ab137043, Abcam). After washing, the membranes were incubated with an HRP-labeled secondary antibody (Sangon, D111042-0100) for 2 h at room temperature and then photographed with chemiluminescent solution.

Fig. 3. The alphabetical labels in brackets within the figure are not clearly explained. These should be defined in the legend for reader clarity.

Response: Thanks for your opinion. The letters in the brackets of the vertical coordinate represent the corresponding functional categories. We added it to the caption of the picture.

Fig. 5. The arrow indicating β-oxidation needs adjustment for clarity. The legend states that orange-colored proteins reflect both expression and modification, but it would be more informative if the figure itself included directional arrows (e.g., up/down) to indicate regulation trends. Consider adding this information directly to the figure.

Response: Thanks for your opinion. As we only conducted qualitative detection on Kbhb and did not obtain quantitative difference information of the modified proteins, we are currently unable to label it as per your suggestion. In the subsequent work, we will conduct in-depth research on this aspect.

The future perspectives section must be elaborated.

Response: Thanks for your opinion. We placed our future perspectives at the end of the discussion. The specific content is as follows:

However, given the complexity of metabolic regulation, the precise mechanisms un-derlying these effects require further investigation. Future studies should focus on elu-cidating the molecular mechanisms by which Kbhb modulates enzyme activity and metabolic pathways. For example, structural studies of Kbhb-modified enzymes could reveal how this modification influences substrate binding, catalytic efficiency, or protein-protein interactions. Additionally, the development of site-specific antibodies or chemical probes for Kbhb would enable more precise functional studies in vivo.

The authors state that "Kbhb occurs on both histone and non-histone proteins," but the discussion is disproportionately focused on histones given the prediction and identification that cytoplasmic proteins are more in abundance. A more balanced analysis or an explanation for the limited insights into non-histone targets would be valuable.

Response: Thanks for your opinion. Although the phenomenon that BHB can serve as a donor for Kbhb is well known, early research on Kbhb was more focused on histones. Based on some recently published literature, we have expanded the content on non-histone Kbhb here:

Qin et al. discovered that the ketogenic diet inhibits glycolysis and tumor cell proliferation by enhancing ALDOB-K108bhb to suppress its enzymatic activity and weaken its binding with fructose-1,6-bisphosphate.43 Aging also increases the level of STAT1-K592bhb, inhibits the interaction between STAT1 and type-I interferon, and leads to a decline in antiviral activity. Inhibiting the level of STAT1kbhb can effectively improve this phenomenon.44 In addition to regulating enzyme activity and protein-protein interactions, Kbhb also affects protein stability. The Kbhb of the transcription factor Snail can prevent the degradation of Snail by blocking the recognition of the E3 ubiquitin ligase FBXL14.45

Update the reference list to include the most recent and relevant literature.

Response: Thanks for your opinion. The latest relevant literature on non-histone Kbhb research has been inserted when responding to the previous opinion.

Reviewer 3 Report

Comments and Suggestions for Authors

In this manuscript, the authors profiled the proteome and metabolome related to beta-hydroxybutyrylation in pig liver and cells.  The methods and results look very good.  The manuscript is well written.  The subject is important, and their results are novel.  I have only minor concerns.

Minor issues:

Line 29: The authors wrote “Cells within the same living organism possess an identical genome.”  This is not always the case.  This sentence should be fixed.

Line 102: The authors wrote “1% protease inhibitor cocktail”.  The specific product name, source, and composition need to be described.  Further down, the authors wrote “For PTM experiments, inhibitors were also added to the lysis buffer, e.g., 3 μM TSA and 50 mM NAM for acetylation, and 1% phosphatase inhibitor for phosphorylation.”.  As above, the specific product name, source, and composition of the “phosphatase inhibitor” needs to be described.  The full names of TSA and NAM need to be written.  In general, a full and clear description of all solutions and procedures needs to be included in the Methods section, and abbreviations need to be defined clearly for readers who are unfamiliar with proteomics experimentation.

Line 108: The source of the BCA kit and antibody needs to be written here.

Line 115: The specific product and source of the trypsin need to be written here.

Line 123: The specific product name of the antibody beads needs to be written here.  The full name of “NETN buffer” needs to be written here.

Line 130: The reversed phase beads need to be described: specific product name, source, bead diameter, pore diameter.  If a trap column was used, this needs to be described.  If heating was used, this needs to be described.

Line 140: The authors wrote “fixed at a mass of 150.0 m/z”.  This is confusing.  This was probably the lower end of their DIA windows.  The authors need to state that they operated the LC-MS in DIA mode and describe their DIA method: range (e.g., 150 - 1000 m/z), number of DIA windows, precursor ion isolation width, DIA window width, and DIA window overlap (if any).

Line 145: The source of the FASTA (e.g., UniProt) needs to be written here.

Line 150: Kbhb is not mentioned as a variable mod.  All modifications must be described here.

Line 151: The authors wrote “FDR was adjusted to <1%”.  The authors need to write if this was performed using Spectronaut or something else.  The authors need to write if this was peptide, protein, and/or PTM site FDR’s.

Line 165: The specific product name, source, and composition of “ultrapure water extraction solution” and “methanol extraction solution” need to be described.

Line 173: The specific product name and source of the protein precipitation plate need to be described.

Lines 174-180: The metabolomics LC-MS and data analysis need more detail.  The LC-MS column type(s), mobile phase compositions, flow rate, ESI polarity and voltage, LC scheduling settings (if used), tandem MS level (e.g., MS/MS/MS), MS resolution and/or scan rate, declustering potential (if applicable), temperature, gas flow rates, collision energy(s), and any other key instrument methods and parameters need to be described in detail.  The “standard solutions with different concentrations” need to be described (e.g., product name and source).  The settings used for their MultiQuant analysis need to be described in detail: integration algorithm and parameters, smoothing settings, regression settings, outlier settings, and any other key methods and settings.

The authors wrote that they shared their proteomics results on iProX, and provided a www address and password; nevertheless, I was unable to access their iProX dataset.  Please provide a logon/password.  Also, a simple table of all of their identified beta-hydroxybutyrylation sites should be provided as supplemental material to their manuscript and/or in their iProX dataset.

The authors have not uploaded their metabolomics data to a public metabolomics data repository.  They need to do this.  If no public metabolomics data repository is suitable for their data, it should be uploaded to a general scientific public data repository such as Harvard Dataverse.  Also, a simple table of all of their identified and/or quantified metabolites versus the biological sample should be provided as supplemental material to their manuscript and/or in their public data repository dataset.

Author Response

Dear Reviewer:

   We are pleased to submit our revised manuscript entitled “Global profiling of protein β-hydroxybutyrylome in porcine liver” (Manuscript ID: biology-3789376) for publication consideration. ​We are extremely grateful for your time and patience in reading our manuscript, which helped us identify many minor issues. We carefully addressed all the issues which you raised. We carefully examined all the details you pointed out to ensure that nothing was missed. We believe that the revised manuscript has been significantly improved.

Our detailed point-by-point responses to each comment are as following:

In this manuscript, the authors profiled the proteome and metabolome related to beta-hydroxybutyrylation in pig liver and cells.  The methods and results look very good.  The manuscript is well written.  The subject is important, and their results are novel.  I have only minor concerns.

Minor issues:

Line 29: The authors wrote “Cells within the same living organism possess an identical genome.”  This is not always the case.  This sentence should be fixed.

Response: Thanks for your opinion. Although the genomes of most somatic cells are the same, there are indeed exceptions. For instance, the gametes (sperm and eggs) produced after meiosis in germ cells have different genomes, with each gamete having a haploid number of chromosomes. Additionally, mutations can occur in somatic cells, such as those caused by environmental factors or replication errors, leading to changes in the genomes of certain cells. Moreover, B cells and T cells in the immune system generate different antibodies and receptors through V(D)J recombination, which also means that their genomes are not entirely identical. Mitochondrial DNA also exhibits heterogeneity, with variations in mitochondrial DNA existing among different cells. Here, we have used a more rigorous expression: Cells within the same living organism generally share an identical genome.

Line 102: The authors wrote “1% protease inhibitor cocktail”.  The specific product name, source, and composition need to be described.  Further down, the authors wrote “For PTM experiments, inhibitors were also added to the lysis buffer, e.g., 3 μM TSA and 50 mM NAM for acetylation, and 1% phosphatase inhibitor for phosphorylation.”.  As above, the specific product name, source, and composition of the “phosphatase inhibitor” needs to be described.  The full names of TSA and NAM need to be written.  In general, a full and clear description of all solutions and procedures needs to be included in the Methods section, and abbreviations need to be defined clearly for readers who are unfamiliar with proteomics experimentation.

Response: Thanks for your opinion. The protease inhibitor mixture contains protease inhibitors III-VII, which we have described in the methods. TSA/NAM and other substances are presented in full name when they first appear. The name of the product of the phosphatase inhibitor (B15002, Selleck) is exactly as this, which contains 8 components, namely water-soluble Sodium Fluoride, Sodium Orthovanadate, Sodium Tartrate, Sodium Molybdate, Imidazole and DMSO-soluble (-)-p-Bromotetramisole oxalate, Cantharidin and Microcystin LR, Microcystis aeruginosa.

Line 108: The source of the BCA kit and antibody needs to be written here.

Response: Thanks for your opinion. We have added the catalog numbers and sources of BCA and antibodies. (L130, L135-136)

Line 115: The specific product and source of the trypsin need to be written here.

Response: Thanks for your opinion. We have added the catalog numbers and sources of trypsin. (Line 145)

Line 123: The specific product name of the antibody beads needs to be written here.  The full name of “NETN buffer” needs to be written here.

Response: Thanks for your opinion. We added the full name of the beads at line 153-154. At line 152, we included the full name of NETN Buffer.

Line 130: The reversed phase beads need to be described: specific product name, source, bead diameter, pore diameter.  If a trap column was used, this needs to be described.  If heating was used, this needs to be described.

Response: Thanks for your opinion. These details are crucial for others to reproduce our results. We had add this part as follow (Line 160-170):

​ The tryptic peptides were dissolved in solvent A and directly loaded onto a homemade reversed-phase analytical column (15 cm length, 100 μm i.d., packed with ReproSil-Pur C18-AQ 1.9 μm resin, 120 Å pore size; Dr. Maisch, Germany #r119.aq). Vanquish Neo UPLC system (ThermoFisher Scientific, America) with trap column (ThermoFisher Scientific, PEPMAP NEO C18 5um 300um×5mm trap 3PK 1500 Bar, #174500, Lithuania) was used. The mobile phase consisted of solvent A (0.1% formic acid in water) and solvent B (0.1% formic acid, 80% acetonitrile/in water). Peptides were separated with the following gradient: 0–0.5 min, 4%B; 0.5–0.6min, 4%–8%B; 0.6–13.6 min, 8%–22.5%B; 13.6–20.5 min, 22.5%–35%B; 20.5–20.9 min, 35%–55%B; 20.9–21.4 min. 55%–99%B; 21.4–22.6 min, 99%B, at a constant flow rate of 400 nL/min on a Vanquish Neo UPLC system (ThermoFisher Scientific, America).

Line 140: The authors wrote “fixed at a mass of 150.0 m/z”.  This is confusing.  This was probably the lower end of their DIA windows.  The authors need to state that they operated the LC-MS in DIA mode and describe their DIA method: range (e.g., 150 - 1000 m/z), number of DIA windows, precursor ion isolation width, DIA window width, and DIA window overlap (if any).

Response: Thanks for your opinion. These details are crucial for others to reproduce our results. We had rewritten this part as follow (Line 171-180):

The separated peptides were analyzed using an Orbitrap Astral instrument oper-ated in data-independent acquisition (DIA) mode with a nano-electrospray ion source. The electrospray voltage applied was 1900 V. DIA parameters were configured as follows: Precursor scan range: 380–980 m/z at 240,000 resolutions. DIA windows: 24 variable windows (400–416, 416–438, 438–466, 466–496, 496–528, 528–562, 562–598, 598–636, 636–676, 676–718, 718–762, 762–808, 808–856, 856–906, 906–946, 946–980 m/z). DIA window width: 2 m/z. Window overlap: 0 m/z. MS/MS scan range: 150–2000 m/z at 80,000 resolution (first mass fixed at 150.0 m/z). HCD fragmentation was performed at 25% nor-malized collision energy. The automatic gain control target was set to 500% with a maximum injection time of 3 ms.

Line 145: The source of the FASTA (e.g., UniProt) needs to be written here.

Response: Thanks for your opinion. The Fasta file is indeed from Uniprot. We have added Uniprot at Line 182.

Line 150: Kbhb is not mentioned as a variable mod.  All modifications must be described here.

Response: Thanks for your opinion. Here are the issues from the previous writing. We have already changed Kla to Kbhb. (Line 187)

Line 151: The authors wrote “FDR was adjusted to <1%”.  The authors need to write if this was performed using Spectronaut or something else.  The authors need to write if this was peptide, protein, and/or PTM site FDR’s.

Response: Thanks for your opinion. We had revised this sentence to: The FDR for protein, peptide and PSM identification in Spectronaut software was set at 1%. (Line 188-189)

Line 165: The specific product name, source, and composition of “ultrapure water extraction solution” and “methanol extraction solution” need to be described.

Response: Thanks for your opinion. We added this information here, the revised sentence is as follow:

The cell pellet was resuspended in 100 μL of ultrapure water (Milli-Q® IQ 7000, Merck Millipore). Then, 50 μL of the suspension was mixed with 200 μL of LC-MS grade methanol (#34860-1L-R, Merck, pre-cooled at -20°C).

Line 173: The specific product name and source of the protein precipitation plate need to be described.

Response: Thanks for your opinion. We added this information here, the revised sentence is as follow:

After passing 180 μL of the supernatant through a protein precipitation plate (#96CD2025-Q-FX, Agela), it was analyzed by Ultra Performance Liquid Chromatography (UPLC) (Waters ACQUITY H - Class D) and Tandem Mass Spectrometry (MS/MS) (QTRAP® 6500+).

Lines 174-180: The metabolomics LC-MS and data analysis need more detail.  The LC-MS column type(s), mobile phase compositions, flow rate, ESI polarity and voltage, LC scheduling settings (if used), tandem MS level (e.g., MS/MS/MS), MS resolution and/or scan rate, declustering potential (if applicable), temperature, gas flow rates, collision energy(s), and any other key instrument methods and parameters need to be described in detail.  The “standard solutions with different concentrations” need to be described (e.g., product name and source).  The settings used for their MultiQuant analysis need to be described in detail: integration algorithm and parameters, smoothing settings, regression settings, outlier settings, and any other key methods and settings.

Response: Thanks for your opinion. We have rewritten this paragraph and supplemented all the information you need. For the MultiQuant analysis settings, we used the default settings of MultiQuant without making any adjustments. All the metabolites in the targeted metabolomics were qualitatively determined with reference to standards, and the integration was performed based on the standards, without involving any special integration parameter settings. The rewritten content is as follows:

The cell pellet was resuspended in 100 μL of ultrapure water (Milli-Q® IQ 7000, Merck Millipore). Then, 50 μL of the suspension was mixed with 200 μL of LC-MS grade methanol (#34860-1L-R, Merck, pre-cooled at -20°C). The mixture was vortexed at 2500 r/min for 2 min. Subsequently, it was rapidly frozen in liquid nitrogen for 5 min, taken out and thawed on ice for 5 min, and then vortexed again for 2 min to ensure homogeneity. This cycle was repeated three times. After centrifugation at 12000 r/min for 10 min at 4°C, 200 μL of the supernatant was transferred to a new centrifuge tube. The tube was placed in a -20°C refrigerator for 30 min and then centrifuged again at 12000 r/min for 10 min at 4°C.

After passing 180 μL of the supernatant through a protein precipitation plate (#96CD2025-Q-FX, Agela), it was analyzed by Ultra Performance Liquid Chromatography (UPLC) (Waters ACQUITY H - Class D) and Tandem Mass Spectrometry (MS/MS) (QTRAP® 6500+).

Liquid Chromatography Conditions:​​ Column: ACQUITY UPLC® BEH Amide (1.7 μm, 2.1 × 100 mm; Waters # 186004801).Mobile Phase: A: ultrapure water (10 mM ammonium acetate, 0.3% ammonia water). B: 90% acetonitrile/water (V/V). Gradient: 0 → 1 .2min, 5:95 A/B (V/V); 8 min, 30:70 A/B (V/V); 9.0→11 min, 50:50 A/B (V/V); 11.1→ 15min, 5:95 A/B (V/V). Flow Rate: 0.4 mL/min. Column Temperature: 40°C. ​Injection Volume: 2 μL.

​Mass Spectrometry Conditions:​​ ​Ionization: ESI with positive/negative polarity switching. Ion Spray Voltage: +5,500 V (POS), -4,500 V (NEG). ​Source Temperature: 500°C. Gas Flow Rates: Curtain Gas 35 psi.

​Quantitative Analysis:​​ Standard Solutions: Calibration curves (NADPH and D-Glucose: 0.1-150000ng/mL, others: 0.01–15000 ng/mL) prepared using ​MS-certified metabolite standards (IROA Technologies #MSML-LEO)​. All standard substances were purchased from Sigma-Aldrich with a purity greater than 99%. Data Processing: Raw data were integrated in ​MultiQuant™ 3.0.3 (Sciex)​​ with default setting.

For the remaining 50 μL of the cell suspension, it was subjected to three cycles of repeated freezing in liquid nitrogen and thawing. After centrifugation at 12000 r/min for 10 min, the supernatant was collected, and the protein concentration was determined by the BCA method. The metabolomic data of the test samples were corrected based on the protein concentrations of the parallel samples.

The authors wrote that they shared their proteomics results on iProX, and provided a www address and password; nevertheless, I was unable to access their iProX dataset.  Please provide a logon/password.  Also, a simple table of all of their identified beta-hydroxybutyrylation sites should be provided as supplemental material to their manuscript and/or in their iProX dataset.

Response: Thanks for your opinion. The previous data link might have expired due to a timeout. We have re-shared the data, and it is valid for 360 days.

URL: https://www.iprox.cn/page/DSV021.html;?url=17552389629927TOL , passwords: zElY

In addition, we have uploaded all the Kbhb sites as supplementary materials as per your request.

The authors have not uploaded their metabolomics data to a public metabolomics data repository.  They need to do this.  If no public metabolomics data repository is suitable for their data, it should be uploaded to a general scientific public data repository such as Harvard Dataverse.  Also, a simple table of all of their identified and/or quantified metabolites versus the biological sample should be provided as supplemental material to their manuscript and/or in their public data repository dataset.

Response: Thanks for your opinion. The raw data of the metabolome has been uploaded to the National Center for Biotechnology Information of China (PRJCA044753).

Round 2

Reviewer 1 Report

Comments and Suggestions for Authors

The authors have addressed all my comments satisfactorily 

Reviewer 2 Report

Comments and Suggestions for Authors

The authors have addressed my queries satisfactorily. The manuscript has improved significantly and can be accepted in its current form.
For Figure 6A, please include the molecular weight markers and specify the type of tubulin used in the blot (e.g., α-tubulin or β-tubulin). Additionally, please ensure that CK is defined or abbreviated appropriately in the figure legend.